# Viral SERPINS—A Family of Highly Potent Immune-Modulating Therapeutic Proteins

**DOI:** 10.3390/biom13091393

**Published:** 2023-09-15

**Authors:** Kyle Varkoly, Roxana Beladi, Mostafa Hamada, Grant McFadden, James Irving, Alexandra R. Lucas

**Affiliations:** 1Department of Internal Medicine, McLaren Macomb Hospital, Michigan State University College of Human Medicine, 1000 Harrington St., Mt Clemens, MI 48043, USA; kylevarkoly@gmail.com; 2Department of Neurological Surgery, Ascension Providence Hospital, Michigan State University College of Human Medicine, 16001 W Nine Mile Rd., Southfield, MI 48075, USA; roxanabeladi@gmail.com; 3College of Medicine, Kansas City University, 1750 Independence Ave, Kansas City, MO 64106, USA; mhamada4@asu.edu; 4Center for Immunotherapy Vaccines and Virotherapy, Biodesign Institute, Arizona State University, 727 E Tyler St., Tempe, AZ 85287, USA; grantmcf@asu.edu; 5UCL Respiratory and the Institute of Structural and Molecular Biology, University College London, 5 University Street, London WC1E 6JF, UK; 6Center for Personalized Diagnostics, Biodesign Institute, Arizona State University, 727 E Tyler St., Tempe, AZ 85287, USA

**Keywords:** serpin, virus, immune-modulating, coagulation, apoptosis, poxvirus, plant virus, baculovirus, biologic

## Abstract

Serine protease inhibitors, SERPINS, are a highly conserved family of proteins that regulate serine proteases in the central coagulation and immune pathways, representing 2–10% of circulating proteins in the blood. Serine proteases form cascades of sequentially activated enzymes that direct thrombosis (clot formation) and thrombolysis (clot dissolution), complement activation in immune responses and also programmed cell death (apoptosis). Virus-derived serpins have co-evolved with mammalian proteases and serpins, developing into highly effective inhibitors of mammalian proteolytic pathways. Through interacting with extracellular and intracellular serine and cysteine proteases, viral serpins provide a new class of highly active virus-derived coagulation-, immune-, and apoptosis-modulating drug candidates. Viral serpins have unique characteristics: (1) function at micrograms per kilogram doses; (2) selectivity in targeting sites of protease activation; (3) minimal side effects at active concentrations; and (4) the demonstrated capacity to be modified, or fine-tuned, for altered protease targeting. To date, the virus-derived serpin class of biologics has proven effective in a wide range of animal models and in one clinical trial in patients with unstable coronary disease. Here, we outline the known viral serpins and review prior studies with viral serpins, considering their potential for application as new sources for immune-, coagulation-, and apoptosis-modulating therapeutics.

## 1. Introduction

### 1.1. Serpins—Serine Protease Inhibitors

Serpins comprise a large family of protease inhibitors, which is ubiquitous in higher eukaryotes, and they have a shared tertiary structure that adopts a unique inhibitory conformation when bound to targeted proteases (Figure 1). Most serpins are *ser*ine *p*rotease *in*hibitors, these are not enzymes but rather are suicide inhibitors of selectively targeted proteases. Currently, over 6000 serpins are known in all kingdoms of life, with 37 found in humans [1,2,3]. They represent roughly 2–10% of proteins in human blood and are the third most common protein family, regulating central pathways in coagulation, apoptosis (cell death), and immune responses, as well as maintaining normal lung, immune and neuronal health [2,3]. Viral serpins have evolved to inhibit selected targets within mammalian coagulation, inflammation, and apoptotic protease-dependent pathways. However, the timing and original genes that have been co-opted by viruses, and whether additional lateral transfer events have occurred among the many kingdoms of life, remains a matter of speculation [2].

Biologically, most serpins function as *ser*ine *p*rotease *in*hibitors, some function as ‘cross-class’ cysteine protease inhibitors, and others as non-inhibitory hormone and protein transporters. The inhibitory mechanism of serpins has been extensively investigated [1,2,3,4,5]. Serpins are folded into a high-energy structure with an exposed region known as the reactive center loop (RCL) that contains a bait recognition sequence which is specific to a target protease. Upon cleavage in the vicinity of this sequence, this energy is expended by incorporating the RCL as an additional β-strand inserted into the central A β-sheet (Figure 1). In this way, serpins utilize some of the energy accumulated during their synthesis to exert a biological function. For inhibitory serpins, the result is a covalent inhibitory serpin–protease complex, in which the activity of both the serpin and the protease is lost, thus often termed suicide inhibition. This mechanism gives rise to an important property of serpins, unlike non-covalent competitive protease inhibitors, they effectively do not have a dissociation rate, and therefore do not need to be maintained at a concentration above a dissociation constant to continue to inhibit the bound pool of proteases. Some serpins, such as the mammalian plasminogen activator inhibitor-1 (PAI-1, SERPINE1) that inhibits tissue- and urokinase-type plasminogen activators (tPA and uPA, respectively), can also exist in a third state where fluidity of the RCL allows partial insertion into the β-sheet A in the absence of cleavage, resulting in a latent, inactive, low-energy state [1,4]. The process of RCL presentation or insertion is, in some cases, modulated by a ligand, such as heparin, which increases the activity of antithrombin III from 100- to 1000-fold (AT, SERPINC1), making it a more potent inhibitor of clotting factors. Similarly vitronectin stabilizes PAI-1 against the latent transition [5].

The impact of serpins as regulatory proteins that maintain normal physiological functions is highlighted by the profound effects of genetic mutations in serpins, which cause a class of disease known as a serpinopathy. These mutations lead, in some cases, to a loss of inhibitory function and, in other cases, a gain of function. In some of these conditions, the deposition of serpin aggregates and protein misfolding is associated with tissue damage and leads to a deficiency state which prevents the effective inhibition of target proteases. For neuroserpin (SERPINI1) and alpha 1 antitrypsin (A1AT, SERPINA1), this aggregation is mediated by a repeating intermolecular contact that generates linear polymers, termed ‘beads-on-a-string’ morphology [6,7]. In A1AT deficiency, polymer formation in the liver causes liver damage (cirrhosis) as well as a lack of effective inhibition of lung neutrophil elastase, causing severe emphysema (lung damage) [8,9]. For neuroserpin (SERPINI1), polymer formation in the central nervous system is the basis for a hereditary dementia termed Familial Encephalopathy with Neuroserpin Inclusion Bodies (FENIB). Notably, the onset and severity of neurodegenerative disease was associated with the rate and magnitude of neuronal protein aggregation [7,10,11].

As polymers share a similar increase in stability to the RCL-inserted, protease-cleaved form of the protein, and polymerization is prevented by pre-incubation with a peptide mimetic of the RCL, it was proposed that polymers form by insertion of the RCL from one serpin into the β-sheet A of an adjacent serpin. This is often referred to as the ‘loop-sheet model’ [1,2,3,4,5]. The feasibility of an alternative mechanism in which polymerization is mediated by a domain swap of the entire 4 kDa C-terminus of the protein has been demonstrated crystallographically using a double cysteine mutant of alpha-1-antitrypsin supported by disulfide trapping experiments [8]. Subsequent structural characterization of polymers extracted from patient livers strongly supports this as the form that is present in vivo [7,9].

Other serpinopathies have been described that are linked to a variety of clinical disorders which are associated with deficiency or dysfunction of the serpin, including antithrombin (AT), which results in excess thrombosis, and C1 esterase inhibitor (C1Inh, SERPING1), which is associated with severe angioedema [2,12,13]. These mutations demonstrate the importance of normal serpin function in maintaining a normal physiological balance in the coagulation, neurologic, and immune pathways.

### 1.2. Serpin Regulation of Coagulation and Immune Responses

Serpins function to regulate cardiovascular, hematological, and immunological responses [Figure 1]. The thrombotic proteases, as components of intrinsic and extrinsic coagulation cascades, produce a series of clotting factor protease activations and fibrin deposition, both of which form clots. These thrombotic cascades have reciprocal interaction and activation with the innate immune response pathways. The thrombolytic enzymes activate plasmin to allow for the breakdown of fibrin in clots, termed fibrinolysis. The thrombolytic proteases can also drive immune cell activation and invasion at sites of endothelial damage, again with the reciprocal activation of immune responses. In particular, the urokinase-type plasminogen activator (uPA), although an activator of plasmin, is considered to be predominantly a meditator of inflammatory response pathways. uPA binds to the uPA receptor (uPAR), allowing local plasminogen activation and forming plasmin that, in turn, activates matrix-degrading enzymes. uPA that is bound to the uPAR activates plasmin, which then activates pro-forms of matrix metalloproteinases (MMPs). MMPs, in turn, break down the extracellular matrix, releasing growth factors and increasing immune cell invasion at sites of tissue injury. The thrombotic and thrombolytic cascades thus have ongoing interactions with immune response pathways, where activated coagulation pathways induce inflammatory cell activation and *vice versa*. This interaction of activated proteases and receptors in coagulation and immune pathways is regulated by serpins, providing opportunities for clinical intervention. For example, heparin, which increases the efficacy of clotting factor inhibition by AT, is one of the most frequently used drugs that is given to patients to reduce excess clotting associated with heart attacks, atrial fibrillation with emboli, pulmonary emboli, and deep venous thrombosis [13].

The capacity of serpins to target a wide range of proteases provides a unique and potentially beneficial method for the use of serpins as therapeutics. Indeed, some serpins are already used as biological therapeutics to treat patients with genetically dysfunctional serpins through augmentation therapy including A1AT and C1Inh. Further, there is currently active development of modified serpins to target excess bleeding in haemophilia. The use of AT as a potential treatment for severe bacterial sepsis with disseminated intravascular coagulation (DIC) has also been assessed [2,13,14,15,16].

Myxomavirus (MyxV) is a poxvirus that infects rabbits, but the MyxV serpin can recognize protease targets outside the rabbit, including proteases in mice and humans. A serpin called Serp-1, that is derived from MyxV, is the most extensively studied virus-derived serpin for use as an anti-inflammatory therapeutic. The areas studied for possible therapeutic benefits are broad due to the potential of serpins for modifying vascular immuno-coagulopathic responses to injury. Preclinical models of disease that have been examined for treatment with viral serpins include vasculitis, unstable angina and myocardial infarction, restenosis after percutaneous coronary intervention, transplant vasculitis and rejection, arthritis, muscular dystrophy, spinal cord injury, wound healing, inflammatory bowel disease, severe colitis, uveitis, diabetic retinopathy, macular degeneration, lupus associated diffuse alveolar lung hemorrhage (DAH), viral acute respiratory distress syndrome (ARDS) with lung infection and coagulopathy, giant cell arteritis, and other acute and chronic inflammatory immune and coagulation disorders [2,17,18,19,20,21,22,23,24,25,26,27,28,29,30,31,32,33,34,35,36,37,38,39,40,41] [Figure 1 and Table 1].

Here, we focus on serpins that are encoded and expressed by viruses and their potential applications as a new class of protein therapeutic. Viral serpins derived from Myxoma poxviruses in particular have been extensively studied in translational and clinical research, with an emphasis on potential therapeutic applications. The current work indicates that these serpins have great promise as potent, anti-inflammatory immune-modulating agents, agents which have demonstrated marked efficacy and minimal adverse effects. The broad efficacy of virus-derived biologics indicates that viral serpins may provide a rich and extensive platform for the discovery of new approaches to treating disease.

**Table 1 biomolecules-13-01393-t001:** Virus-derived Serpins–Prior Pre-clinical and Clinical analyses of Virus-derived Serpins as Therapeutics for Inflammatory Disease. Diseases that have been assessed and efficacy are listed.

Viral Serpin	Inflammatory Disorder	Target/Outcomes	Subjects Studied	Reference
Serp-1	Atherosclerotic Plaque Acute coronary syndromes with stent implant	(1) Phase II Clinical Trial, Mechanism extensively studied/Reduced markers of cardiac damage, MACE = 0, No neutralizing antibodies (2) Preclinical–Reduced intimal hyperplasia	(1) Human Clinical Trial, Randomized dose escalating trial at 7 sites Canada and US (2) Preclinical–Rabbits–Reduced intimal hyperplasia	[20,24]
	Angioplasty injury and intimal plaque restenosis	Efficacy and Mechanism extensively studied	Rabbits, Rats, Mice, Swine/Reduced intimal hyperplasia and inflammation in rodents and rabbits	[20,21,22]
	Aortic Allograft Transplant	Established uPAR as central for Serp-1 therapeutic efficacy	Mouse and rat	[21,33,42]
	MHV68—Mouse Herpes virus aortitis and lung inflammation	Serp-1 reduced mortality, whereas NSP did not Reduction in monocytes and pro-inflammatory cytokines, reduced lung hemorrhage, NSP not effective	Mice	[26,43]
	Carotid compression—Atherosclerotic Plaque	Reduction in carotid arterial inflammation and plaque	Mice	[22]
	Giant Cell Arteritis Human Temporal artery biopsy implants	Reduction in monocytes and pro-inflammatory cytokines	SCID Mice	[25]
	Ebola infection	Reduced liver damage and improved survival	Mice	[26]
	Diffuse Alveolar Hemorrhage (DAH) in Pristane induced Systemic Lupus Erythematosus	Reduced uPAR and Complement and reduced macrophage infiltrates	Mice	[17,44]
	SARS-CoV-2	Reduction in M1 macrophage recruitment in cardiac and pulmonary tissue	Mouse adapted SARS models in C57Bl/6 and BALB/c mice	[39]
	Colitis model	Serp-1 reduced mortality	Mice	In preparation
	Collagen induced arthritis	Reduced joint swelling without antibodies to Serp-1 treatment	Rats	[28]
	Retinal Inflammation—Uveitis Corneal wound healing	Retinal-Intra-vitreal injection of AAV vector expressing Serp-1 Corneal—topical	Mice	[29,45]
	Periodontal bacteria with associated increased atherosclerotic plaque	Decreased pro-inflammatory markers and reduced plaque	Mice	[30]
	Scar Reduction in Wound Healing	Accelerated wound healing-Improved collagen formation in wound bed Effects blocked by uPAR antibody	Mice	[31]
	Transplant Rejection	Rat and Mouse renal, aortic and heterotopic heart allografts Rat to mouse cardiac xenografts	Human, Rat, Mice	[32,33,46,47]
	Angiogenesis	Effects blocked by uPAR antibody in wound healing Reduced availability of VEGF	Chicken angiogenesis CAM Model Mouse wound healing model	[31,34]
	Pancreatic Cancer subcutaneous implants	Reduced tumor weight for pancreatic cell line implant and reduced macrophage infiltration	Human to Mice cell xenografts in SCID mice	[35]
	Spinal Cord Injury Balloon angioplasty crush injury	Improved motor function, reduced inflammation and improved neuronal growth Local infusion	Rats Local infusion	[36,37]
	Duchenne Muscular Dystrophy	Reduced leukocyte invasion, Improved myofibril organization Serp-1, PEGSerp-1	Double knock out DMD Mice	[38]
Serp-2	Angioplasty injury and intimal hyperplasia	Reduced inflammation and intimal hyperplasia	Mice	[23,40]
	Carotid cuff compression injury in hyperlipidemic mouse models	Reduction in aortic Atherosclerotic Plaque Development	ApoE^null^ Mice carotid cuff compression	[40]
	Aortic Transplant	Reduced intimal plaque and inflammation Inhibition of Granzyme-B mediated apoptosis	Mouse aortic Allografts (1) WT C57Bl/6 to BALB/c and (2) Granzyme B KO donor allografts	[40]
	Liver Transplant	Improved survival, reduced hepatocyte necrosis	Mice	[41]
SPI-1	Chemotherapy	Multi-pathway inhibitor of apoptosis	In vitro	[48]
	Viral Host Defense and Vaccination		Mice, in vitro	[48]
SPI-2 and CrmA	Chemotherapy	Multi-pathway inhibitor of apoptosis	In vitro	[39,40,48]
	Neurodegeneration		Mice, in vitro	[49]
	Fulminant Liver Failure		Mice, in vitro	[50,51]
	Autoimmune Hepatitis	Reduction in inflammatory cell (CD11), inhibitor of apoptosis	Mice, in vitro	[50,51]
	Ischemic-Reperfusion Injury/Chemotherapeutic Cardiotoxicity	Multi-pathway inhibitor of apoptosis	Mice, in vitro	[52,53]
SPI-3	Hemophilia	Inhibits uPA, plasmin, and tPA	Mice, in vitro–proposed function	[54,55]

## 2. Virus-Derived Serpins

Viral serpins operate as extracellular and intracellular agents, both of which have demonstrated efficacy in animal models, exerting effects on immunologic, thrombotic, and apoptotic processes (Figure 2). These virus-derived proteins provide potential new treatments with advantages that include: (1) efficacy at very low doses (micrograms of drug per kilogram of patient), (2) the ability to target sites of protease activation, e.g., focused efficacy, (3) minimal side effects and low antigenicity at administered doses, and (4) the capacity for modification to alter protease target specificity [2].

The known viral serpins that have been the most extensively studied are derived from poxviridae, but serpin sequences are also reported in herpesviruses and insect viruses. The majority of viral serpin therapeutic studies have been performed with myxomavirus, vaccinia, or cowpox viral serpins. Myxomavirus is a poxvirus that is lethal in European rabbits and was introduced to cull the rabbit population in Australia. Vaccinia virus is a vaccine strain that is used for developing vaccines for smallpox virus and, more recently, as an oncolytic virus for cancer treatments. Vaccinia was originally thought to be derived from cowpox, but more recent work suggests that the vaccinia virus is most likely to have been derived from a virus we now call horsepox. These virus-derived serpins have been studied as potential therapeutics in isolation from other viral components, as described in the following sections.

### 2.1. Poxvirus Serpins

Myxomavirus encodes and expresses two distinct serpins, Serp-1, which is secreted by an infected cell, and Serp-2, which operates intracellularly, albeit displaying extracellular anti-inflammatory activities. Serp-1 binds and inhibits extracellular thrombolytic and thrombotic proteases as well as complement proteases [17,18,19,56,57], while Serp-2 inhibits intracellular and cell surface apoptosis pathway proteases, granzyme B, a serine protease, and caspases 1 and 8, which are cysteine proteases [23,29,40,41]. CrmA and Spi2 are intracellular serpins derived from cowpox and vaccinia virus and also target the apoptotic pathways. Other poxviral serpins include the Spi1 and Spi3 serpins, both of which are intracellular [48].

2-1A).Serp-1

Serp-1 has been extensively investigated over the past 30 years as a new class of virus-derived immune-modulating therapeutic. Serp-1 has developed a high efficacy during evolution, protecting MyxV from attack and clearance by host inflammatory cells at picomolar concentrations when secreted by virus-infected cells. When the Serp-1 gene is deleted in myxomavirus, this lethal European rabbit infection becomes benign and the blockade of rabbit immune cell responses, that are produced in response to myxoma infection, is lost [56,57]. Purified Serp-1 protein has proven effective as a therapeutic at reducing damaging inflammation in a wide array of animal models of disease when delivered systemically at picogram to microgram doses (Figure 1 and Figure 2 and Table 1), and shows promise as a potential treatment for other disorders [17,18,19,20,21,22,23,24,25,26,27,28,29,30,31,32,33,34,35,36,37,38,39,40].

Serp-1 regulates the excess activation of immune responses through the inhibition of coagulation and complement proteases. Serp-1 binds thrombolytic and thrombotic proteases (tPA, uPA, plasmin, fXa, and thrombin (in the presence of heparin, as well as complement proteases, as demonstrated by immunoprecipitation and mass spectrometry [17,57]. Serp-1, as in the case for the mammalian serpins PAI-1, alpha 2 anti-plasmin (A2AP), protease nexin-1 (PN-1), and A1AT, binds and inhibits more than one serine protease. Serp-1 modifies leukocyte adhesion, activation, gene expression, and calcium homeostasis [18,19]. Additionally, Serp-1 has potent anti-inflammatory activity through its inhibition of the uPA/uPA receptor (uPA/uPAR) and complement receptors, and associated effects on monocyte and endothelial cell responses [18,19,21,58].

In the following sections, we will review preclinical and clinical studies of diseases for which Serp-1 treatment has been examined which have supported its therapeutic safety and efficacy (Figure 2).

2.1A)-1.Acute and Chronic Inflammatory Diseases2.1A)-1-1.Atherosclerotic Disease2.1A)-1-1-1.Percutaneous Coronary Intervention (PCI) and Restenosis in Unstable Atherosclerotic Plaque—preclinical testing

Inflammatory macrophage- and T-cell-mediated atherosclerotic plaque rupture induces the exposure of the inner atheroma layers, specifically fat (lipids) and collagen layers, and this in turn causes platelet activation and local thrombosis. Inflammatory plaque rupture in coronary arteries and carotid and/or cerebral arteries is associated with acute thrombotic arterial occlusion causing myocardial infarction (MI) and cerebrovascular accidents (CVA or stroke). Percutaneous coronary intervention (PCI) also can cause endothelial damage, dissection and acute thrombosis or more simple intimal hyperplasia with restenosis and/ or thrombosis [20,24].

Purified Serp-1 protein was first investigated as a potential therapeutic in a hyperlipidemic rabbit model for the accelerated atherosclerotic plaque that is seen after angioplasty, termed restenosis. Serp-1 is a secreted 55kDa glycoprotein that binds and inhibits tissue and urokinase-type plasminogen activator (tPA, uPA), plasmin, factor X, and thrombin as well as complement proteases. Single picogram to nanogram doses were given by local infusions (Wolinsky perforated balloon catheters) immediately after angioplasty. Significantly reduced intimal hyperplasia and inflammation were demonstrated at 30-days follow-up. Subsequent studies examined systemic infusions and stent implants [20]. This was the first report on the use of a viral anti-inflammatory serpin as an immune-modulating treatment. At low-dose infusions, only the primary site demonstrated plaque improvement, whereas, at high-dose infusions, more systemic reductions in plaque development were detected. Associated with the reduced plaque after treatment, there was reduced mononuclear cell infiltration in the arterial intimal layer when compared to control saline or inactive Serp-1 SAA (a Serp-1 protein where the Arg and Asn at the P1-P1′ RCL scissile bond are replaced by Ala). Overall, the marked reduction at 30-days follow-up after single picogram doses were given at the time of angioplasty injury demonstrated marked efficacy. No adverse effects were seen in this study. PCI with stent implant was also assessed in rabbits, and demonstrated efficacy with preventative treatments given for three days post stent implant. This study demonstrated early promise for Serp-1 as a virus-derived immune modulating protein, potentially over other current standard therapeutics that are given after PCI, where restenosis can occur in up to 30% of lesions [20,24].

Serp-1 further demonstrated promise in angioplasty and aortic transplant models in rodent models. One study investigated the effects of Serp-1 following rat iliofemoral angioplasty and injury. A reduction in intimal hyperplasia was seen in animals that were treated with wild-type Serp-1 when compared to treatment with a range of Serp-1 reactive center loop mutants [21]. Inflammatory plaque formation was also examined in mouse aortic allograft transplants. In this transplant vasculopathy model, Serp-1 treatment improved inflammation and intimal hyperplasia. Serp-1 blocked intimal hyperplasia after aortic allograft transplant in donor aortic transplants derived from PAI-1-deficient aortic donor allografts (C57Bl/6 to BALB/c mice) [21]. However, Serp-1 treatment was ineffective after donor aortic transplants in uPAR-deficient mouse aortic donor allografts. Serp-1, but not a reactive center loop mutant, up-regulated PAI-1 serpin expression in human endothelial cells. The treatment of endothelial cells with an inactivating antibody to uPAR and vitronectin blocked these observed Serp-1-induced changes. This indicated that Serp-1 treatment is not dependent upon mammalian PAI-1 expression. These early studies supported the idea that Serp-1 reduced vascular inflammatory responses to injury through native uPA/uPAR receptor interactions and potentially via uPAR lipid raft protein interactions, as with vitronectin [19,20,21].

In experiments by Ilze Bot and Erik Biessen, Serp-1 treatment was investigated in a hyperlipidemic ApoE^null^ mouse model after carotid cuff compression injury. Continuous infusions of Serp-1 by osmotic pump over 30 days, again, significantly reduced arterial (carotid) inflammation and plaque growth at the site of carotid compression. This study further demonstrated safety and efficacy for continuous infusions of Serp-1 [22].

To better elucidate the mechanisms behind Serp-1-mediated decreases in arterial plaque growth, the signaling pathways were investigated. In vitro analysis of Serp-1 revealed decreased monocyte activation, as well as decreased membrane fluidity. In these studies, Serp-1 was noted to bind to the uPA receptor (uPAR) and to increase beta actin (filamin expression) expression and decrease CD18 (beta integrin) expression [18,19]. Decreased membrane fluidity is seen in a variety of disease states including hypercholesterolemia, hyperlipidemia, and atherosclerotic heart disease. Both Serp-1 and Serp-2 decreased the expression of WNT2 as well, which is an important signaling molecule in cell-to-cell communication. Decreased expression of the atherogenic molecules CD5, SELE, and VCAM1 was also seen when monocytes were treated with Serp-1, providing another molecular pathway for atherosclerotic plaque reduction with this therapeutic [23].

2.1A)-1-1-2.Periodontal Disease and Associated Atherosclerotic Plaque Development

The oral microbiome is closely associated with atherosclerosis and heart health. In a mouse study with aortic injury and balloon angioplasty, mice infected with *Porphyromonas gingivalis* had increased intimal plaque following balloon angioplasty. When treated with Serp-1, reductions in this coronary intimal plaque were observed. Toll-like receptor 4 and myeloid differentiation primary response 88 were also decreased following Serp-1 treatment, suggesting a potential anti-inflammatory mechanism for Serp-1 treatment [30].

2.1A)-1-1-3.Acute Coronary Syndrome (ACS)—Clinical Trial

Based on the prior preclinical studies which demonstrated Serp-1-mediated reductions in atheroma after aortic and carotid injury, and following a safety assessment in rat and primate models, Serp-1 was investigated in a randomized, blinded, dose-escalating study in patients with unstable coronary syndromes receiving coronary stent implants [24]. Acute or unstable coronary syndrome (ACS) is caused by inflammatory macrophage infiltration into artherogenic plaque in the coronary arterial intimal layer that leads to eventual plaque rupture and the superimposed activation of thrombosis. This causes either total arterial thrombotic occlusion with heart attack, stroke, or peripheral vascular occlusion or intermittent occlusion termed unstable angina (also termed non ST elevation myocardial infarction or NSTEMI). In the Phase 2A trial that was undertaken at seven sites in the US and Canada, Serp-1 was infused at doses of 0, 5, and 15 µg/kg daily for three days by intravenous bolus together with standard of care and with comparison to patients receiving only standard of care with no Serp-1 protein treatment. Treatment with the higher dose of Serp-1 significantly reduced troponin I and CK-MB, markers of heart damage, when given immediately after coronary stent implant in patients with unstable angina and non-ST elevation MI, (i.e., in patients with ACS). There were no adverse events, and the major adverse cardiovascular event (MACE) rate of death, myocardial infarction, or coronary revascularization at the higher dose of Serp-1 was zero; further, no neutralizing antibodies were detected. There was no significant decrease in plaque growth seen upon conducting a follow-up intravascular ultrasound. The reduction in myocardial damage supported the further assessment of Serp-1 in ACS patients undergoing stent deployment [24].

2.1A)-1-2.Inflammatory Vasculitis Syndromes—Giant Cell Arteritis and Takayasu Disease

Given the efficacy of Serp-1 treatment in atherosclerotic plaque and intimal hyperplasia, Serp-1 treatment was also examined in models for inflammatory vascular disease.

2.1A)-1-2-1.Giant Cell Arteritis

Giant cell arteritis and Takayasu’s disease are two large-vessel inflammatory vasculitic syndromes (IVS). IVS are associated with sudden onset blindness, aneurysm formation, and diffuse vascular occlusions. High-dose pulsed steroids, the chemotherapeutic cyclophosphamide, aspirin, and steroid-sparing drugs such as the interleukin 6 receptor (IL 6R) inhibitor Tocilizumab are currently the mainstays of treatment for IVS. However recurrent disease is frequent and additional therapies are needed. Temporal artery (TA) biopsies from human patients with suspected giant cell arteritis (GCA) were obtained for the diagnosis of temporal arteritis and GCA. In a blinded analysis, full-thickness sections of the TA biopsy specimens were implanted into the aorta of immunodeficient SCID mice. The mice implanted with human TA biopsy specimens that were diagnosed as positive for giant cell arteritis had significantly more inflammation than those with negative implants, confirming the model. Peripheral blood mononuclear cells were infused to more closely simulate the inflammation in GCA. Serp-1 reduced inflammatory cell invasion, including a reduction in Th1, Th17, T-reg, and, more importantly, the cytokine interleukin 1-β (IL-1β). This study demonstrated that unmodified Serp-1 may provide a new immune-modulating biologic for the treatment of IVS where current treatments have failed, with limited side effects [25].

2.1A)-1-2-2.Mouse Gamma Herpes Viral Infection (MHV68)

Serp-1 treatment was also assessed in a mouse gamma herpesvirus herpesvirus (MHV68) model for inflammatory vasculitis. Vascular inflammation is seen in some severe viral infections, as has been reported in the recent COVID19 pandemic [39]. Sepsis can induce a severe imbalance in the coagulation pathways wherein excess clot formation in bacterial and viral sepsis can lead to the depletion of proteases, inducing thrombosis. The depletion of coagulation factors can then cause subsequent bleeding. This imbalance in the clotting and clot-dissolving cascades is termed DIC. In severe bacterial, fungal, or viral infections with sepsis, treatments are often ineffective and have a high rate of mortality. In prior work, AT and heparin treatments in bacterial sepsis with DIC were assessed. While early studies showed some promise, a larger study indicated only an equivocal benefit [14,15]. Given the capacity for Serp-1 to inhibit thrombotic, thrombolytic, and complement proteasesa, and that it functions as an inhibitor that reduces intimal plaque and arterial inflammation, Serp-1 treatment was postulated to hone to sites of protease activation in MHV68 virus-induced inflammatory vascular disease, sepsis, and DIC. Serp-1 treatment was compared with neuroserpin (NSP) treatment in this model, as NSP inhibits only the thrombolytic proteases tPA and uPA, and does not bind or inhibit fXa or thrombin. In this study, Serp-1 improved survival, with 60% of mice surviving at day 150, versus 0% in the saline-treated and NSP-treated mice. Serp-1 reduced the viral load, lung hemorrhage, and aortic and lung inflammation [26]. Additionally, NSP suppressed splenic cell responses during MHV68 infection, while Serp-1 increased the amount of CD11c+ monocytes and dendritic cells and reduced the amount of resident tissue macrophages. Serp-1 treatment also modulated the gene expression for selected coagulation and inflammatory responses when compared to NSP. Beneficial effects of Serp-1 were also seen in mouse-adapted Zaire ebolavirus in wild-type BALB/c mice, with improved survival and reduced tissue necrosis. In recent work with a mouse-adapted SARS coronavirus infection, treatment with PEGylated Serp-1 (PEGSerp-1), again, significantly improved clinical scores and outcomes, with reduced lung consolidation and inflammation [39]. Thus, in both DNA and RNA viruses, Serp-1 has demonstrated therapeutic efficacy in mice models for deadly viral infections [26].

The treatment of MHV-68 infections in interferon gamma receptor knockout mice was further tested as a model for lethal vasculitis, DIC, colitis, and lung hemorrhage in a study investigating the gut microbiome. The suppression of gut bacteria by the oral administration of an antibiotic cocktail worsened MHV-68 infection and blocked Serp-1 treatment efficacy, indicating that the Serp-1 therapeutic efficacy in this MHV68 model was dependent upon an intact gut microbiota [27].

2.1A)-1-3.Autoimmune Disorders2.1A)-1-3-1.Transplant Rejection

While acute transplant rejection is well-controlled with current immunosuppressants, chronic transplant vasculopathy and rejection remain a barrier to long-term transplant function. Vascular rejection is closely associated with chronic transplant vascular inflammation, occlusions, rejection, and scarring. Acute and hyperacute rejection also limit allograft transplant outcomes as well as function and survival after xenotransplant. A reduction in chronic rejection and vasculitis as well as the improvement of xenograft transplant outcomes would be paradigm-shifting in improving long-term outcomes, particularly given the limited supply of organs that are available for transplantation. There are, however, barriers which persist, from immunologic to physiological barriers, not to mention zoonosis and ethical considerations in developing functional xenotransplants [21,32,33].

Given the efficacy of Serp-1 treatment in reducing vascular inflammation and in atherosclerosis, Serp-1 treatment was assessed in rat and mouse aortic and rat renal allograft transplant models for transplant rejection and transplant vasculitis. Serp-1 or control infusions were given by venous bolus infusions immediately following aortic allograft transplant. Both intimal plaque and early macrophage, and lymphocyte invasion in the intimal, medial, and adventitial layers, were reduced with Serp-1, and it also attenuated the depletion of medial smooth muscle cells [21]. Zhong and Wang demonstrated that vascular inflammation and organ scarring were significantly reduced in renal allografts [46]. Serp-1 also was effective at improving cardiac heterotopic allografts in mice and rats to mouse xenotransplants. However, the survival of the xenografts was not prolonged [32,47].

In the mouse aortic allograft transplant model, the role of the uPA/ uPAR complex and the mammalian PAI-1 serpin in Serp-1-mediated reductions in vasculitis were analyzed using donor-transplanted aortic segments from mouse models that were deficient in uPAR or PAI-1 knock out mice. Serp-1 blocked plaque growth after aortic isograft transplant and after wire-induced injury in PAI-1-deficient mice, indicating that PAI-1 expression is not required for Serp-1 to block the development of vasculopathy. However, Serp-1 did not inhibit plaque growth in uPAR-deficient aortic allografts, indicating that Serp-1 required uPAR expression in the donor graft to reduce aortic allograft inflammation and intimal hyperplasia [21].

2.1A)-1-3-2.Rheumatoid Arthritis

Rheumatoid arthritis (RA) is a severe and chronic autoimmune and inflammatory joint disease, one of the most common in the world. Although the complex inflammatory pathways of RA are not completely understood, the rodent collagen-induced arthritis model has been well-established to study antigen-induced autoimmune-based disease. Rats that were administered Serp-1 at 50 µg/kg via intravenous (IV) injections had reduced clinical arthritis, with reduced joint swelling when it was given at the time of inducing the disease [28]. The clinical severity was significantly lower and bony erosions on blinded radiographs were also reduced. However, the treatment was less effective when given later, after initiating joint inflammation. Delayed-type hypersensitivity reactions were lower in the Serp-1 treatment group, while no antibodies to Type II collagen were detected. Minimal histopathological synovial changes were detectable in the recipient rats and no neutralizing antibodies to Serp-1 were developed. These results indicate promise for a therapeutic potential of Serp-1 in arthritis [28].

2.1A)-1-4.Systemic Lupus Erythematosus and Diffuse Alveolar Hemorrhage (DAH)

Diffuse alveolar hemorrhage (DAH) is a rare and highly lethal complication in systemic lupus erythematosus (SLE), occurring in 1–5% of cases but with a mortality up to 50–80%. Current treatments are limited, although some newer treatment approaches such as infusion of the clotting factor VII are under investigation [17,44]. WT Serp-1 and a PEGylated variant, PEGSerp-1, that has an increased half-life from 20 min to 8 h, were assessed for treatment in a mouse model of pristane-induced DAH. PEGSerp-1 markedly reduced DAH in the pristane model in the mouse DAH models, with significant associated reductions in macrophage invasion and detectable uPAR and complement membrane attack complex, and reduced macrophage invasion, as determined by immunohistochemical analysis (IHC). Of even greater interest, both WT Serp-1 and PEGSerp-1 were effective when given prophylactically immediately after a pristane injection was given to induce DAH [17,44]. However, PEGSerp-1 also significantly improved pristane-induced DAH when given as a delayed treatment, starting seven days after inducing DAH. The capacity to improve outcomes in the SLE DAH model would be of greater potential benefit in a clinical setting [17].

2.1A)-1-5.Neurological, Musculoskeletal and Ophthalmological Disorders2.1A)-1-5-1.Spinal Cord Injury

Spinal cord injury (SCI) initiates a severe pro-inflammatory response in the dura, resulting in a debilitating loss of function as well as increased mortality. The ensuing hemorrhage and necrosis from the initial injury is complex with associated severe inflammatory destruction. Kwiecien et al. assessed Serp-1 infusion in a balloon crush injury model of the distal thoracic spine in a rat model. Thirty-three mature male rats sustained an epidural crush SCI and then received a subdural infusion of Serp-1 or a second Myxomavirus-derived chemokine modulating protein, M-T7, in comparison to steroid infusion (dexamethasone, DEX) or saline [21,33]. The infusions were given locally at the site of SCI for seven days via osmotic pump, with a separate group of rats receiving intraperitoneal infusion. The clinical endpoint monitoring included bodyweight, hemorrhagic cystitis, and bilateral pinch response. The spinal cord was sectioned and the macrophage infiltrates were measured at the site of crush injury. Rats infused with DEX had side effects of DEX toxicity, including dermal atrophy and body weight loss. The rats infused with Serp-1 and M-T7 had no such toxicity. Serp-1 improved the clinical bilateral pinch withdrawal responses. The subdural infusions of all treatments had reduced CD68+ macrophage numbers. This study indicates Serp-1′s efficacy in a rat SCI model [36].

The effects of Serp-1 on rat SCI were validated in an additional study on 58 rats which determined the optimal dose (0.2 mg/week) with comparison to a saline control. Motor function was improved with Serp-1 infusion, together with reduced phagocytic macrophages at the site of SCI injury in rats infused with from 8 µg to 200 µg of Serp-1. Large amounts of myelin-rich necrotic debris and red blood cells were detected between infiltrating macrophages, supporting the inhibition of active macrophage phagocytosis. The macrophage counts in the Serp-1-treated rats were reduced to approximately half the macrophage counts in the saline-treated rats at eight weeks, supporting an anti-inflammatory effect of Serp-1 treatment for SC crush injuries [37].

2.1A)-1-5-2.Duchenne Muscular Dystrophy

Duchene muscular dystrophy (DMD) is an x-linked recessive disease which causes progressive weakness and muscle wasting in early childhood. Over 1000 variants of DMD have been reported. Newer gene therapy approaches have been developed for severe DMD variants but they are limited to a small percentage of genetic mutations. However, inflammation and fibrosis cause ongoing severe diaphragm and cardiac muscle damage with respiratory and cardiac failure. Treatment is often limited to high-dose steroids with limited efficacy and associated side effects. Studies which investigated wild type Serp-1 and PEG Serp-1 treatment demonstrated a reduction in M1 macrophages with reduced diaphragm fibrosis and increased myofibril diameters [38]. Thus, PEGSerp-1 has the potential to be used as a therapeutic approach in DMD pathology and potentially improve muscle regeneration [38]. Studies are in progress to assess AAVSerp-1 as a gene therapy approach to provide a sustained benefit with reduced damaging inflammation in DMD patients.

2.1A)-1-5-3.Retinal Inflammation

Retinal inflammation is associated with vision loss in uveitis, age-related macular degeneration, and diabetic retinopathy. As a new approach to treatment, Lewin, Ildefonso et al. developed an AAV2 expression vector. AAVSerp-1, with expression confirmed by Western blot, was used to prevent endotoxin-induced uveitis in a mouse model. In a pilot study, AAV2 Serp-1 reduced endotoxin-induced uveitis in mouse models. Corneal injury was also examined by Zhu et al., who demonstrated improved healing with Serp-1 given topically together with IP injections. This allows for future studies investigating intra-vitreal injection of this AAV vector in vivo with potential clinical applications in ocular inflammatory diseases or as a treatment approach for chronic disorders such as DMD [29].

2.1A)-1-6.Wound Healing

Serp-1 has been investigated in early wound healing models. The chronic impairment of wound healing is a major health problem in diabetics, with marked increases in risk for morbidity with infection, amputation, and mortality. Serp-1 was applied topically at low doses or via release from a chitosan-collagen hydrogel at sites of 3.5 mm punch biopsy after 15 days of treatment in a mouse model [31]. Topical Serp-1 treatment significantly accelerated wound healing, with repeated dosing at a lower dose proving more effective than single high dosing. Continuous low-level Serp-1 release from the chitosan collagen hydrogel similarly improved wound healing. These effects were blocked by uPAR antibody, again indicating the uPA/uPAR as a key site for Serp-1-mediated anti-inflammatory activity. The Serp-1-treated wounds had elevated arginase-1, expressing M2-polarized macrophages, which indicated a relative increase in anti-inflammatory M2 when compared to pro-inflammatory M1 responses, with associated periwound angiogenesis. Improved collagen maturation and organization were also demonstrated with more normal skin architecture at the wound site. Serp-1 was deemed as a potential therapeutic for reducing scarring in deep wounds [31]. Topical Serp-1 treatment also proved effective at improving wound healing after alkali-induced corneal injury in mice [29,45]

2.1A)-1-7.Cancer Therapeutics

Angiogenesis is a critical factor in the development of a broad range of chronic diseases such as malignant tumors, as well as a natural defense against vascular occlusions in arthritis, wound healing, and cardiovascular disease. Myxomavirus is also under investigation as an oncolytic virotherapy for solid tumors. To assess the potential for Serp-1 to reduce the blood supply to tumors, Serp-1 was initially investigated by Richardson and Hatton in an angiogenic chicken chorioallantoic membrane model (CAM). Serp-1 inhibited endogenous angiogenesis in a dose-dependent fashion through significantly inhibiting gene expression of laminin and reducing the expression of VEGF. Serp-1 did not affect the CAM following the rapid growth phase. This was not seen after treatment with the Serp-1SAA reactive center loop mutant that lacks inhibitory activity for tPA, uPA, plasmin, and fX. This study highlighted how Serp-1 modulates the angiogenic process by specifically targeting endothelial cells and reducing the availability of VEGF [34].

Serp-1 treatment was also assessed in a model for subcutaneous implant of pancreatic tumor cells in mice. Inflammation in pancreatic cancer is associated with tumor-associated macrophage and myeloid-derived suppressor cell (MDSC) activity, which correlates with cancer progression. Human cancer cells were engrafted subcutaneously into SCID mice, and were subsequently treated with Serp-1, NSP, or M-T7. Serp-1 and NSP both inhibited the growth of the pancreatic cell line Hs776t four weeks following implantation. Serp-1 also inhibited the growth of a second pancreatic cell line, MIA PaCa-2. The inhibition of tumor growth by Serp-1 was associated with a significant decrease in splenocyte MDSC counts as assessed by flow cytometry, without reductions in other splenocyte subpopulations. Serp-1 and NSP both also reduced tumor-associated macrophage infiltration [35]. This study suggested that Serp-1 may contribute to the oncolytic tumor suppressing activity of myxomavirus.

2.1A)-1-8.Severe Acute Respiratory Distress syndromes, SARS-CoV-2 Infection

Severe acute respiratory distress syndrome (ARDS) during coronavirus-2 (SARS-CoV-2) infection is associated with cytokine storm, uncontrolled inflammation, and systemic thrombosis with a high mortality in both those with and without significant comorbidities. Mortality is as high as 25–40% once in the ICU setting. Antivirals have been used clinically in early COVID-19 infection to reduce the infection severity in the early phase of viral infection; however, treatments are limited for the later immune-thrombotic syndromes and cytokine storm. In a prior work with the MHV68 mouse herpesviral infections, lung hemorrhage as well as vasculitis were improved with Serp-1 treatments [26]. To this end, purified PEGSerp-1 was tested in two mouse-adapted SARS-CoV-2 models: the MA10 SARS-CoV-2 infection in BALB/c mice and the MA30 SARS-CoV-2 infection in C57Bl/6 mice [39]. PEGSerp-1 given prophylactically at the time of the initial infections significantly reduced lung consolidation, with associated reductions in M1 macrophage invasion in both the lung and heart. When given as a delayed treatment in the MA30SARS infection model, Serp-1 again improved the outcomes at a lower dose. Detectable uPAR and complement membrane attack complex (MAC) were closely associated with reductions in clinical symptoms, weight loss lung consolidation, and macrophage invasion in both the MA10 and MA30 models. Of interest, the detectable uPAR on IHC staining was significantly reduced in effective PEGSerp-1 doses with delayed treatments while C5b9 complement membrane attack complex (MAC) detection was reduced at all Serp-1 doses. This would suggest, again, a central role of the uPAR in Serp-1-mediated immune modulation. This study with PEGSerp-1 treatment in mouse models of SARS-CoV-2 provides another example of improved outcomes in mouse models of severe vasculitis and lung infection, with therapeutic potential for treatment during the later phases of viral infections [39].

2-1B).Serp-2 and CrmA

Myxomavirus Serp-2 is an intracellular cross-class serpin with distinct functions that differ from the Serp-1 protein discussed above. Cytokine response modifier A (CrmA) is also a cross-class serpin that is expressed by cowpox virus. CrmA, also termed Spi2, shares similar cross-class inhibitory functions to Serp-2 [40,41]. Both block the serine protease granzyme B as well as the cysteine proteases caspase 1 (interleukin converting enzyme) and caspase 8 (Figure 2). Granzyme B and caspases 1 and 8 drive apoptosis, a form of programmed cell death that cells can use to limit viral replication: a form of auto-destruct sequence. Viral anti-apoptotic functions are projected to protect viruses, allowing viral replication to proceed without inducing cell death [40,41].

In vitro, Serp-2 and CrmA both significantly reduce camptothecin (CPT)-induced elevations in caspase 3 and caspase 7 in monocytes. However, only Serp-2 reduces caspase 3 activity in Jurkat T-cells when compared to controls. In T-cells treated with cytotoxic T-cell medium (CTLm), Serp-2 attenuated CTL-mediated increases in Granzyme B/ Caspase 8 and Caspase 3/7. CrmA did not attenuate Granzyme B/Caspase 8 and Caspase 3/7 expression. Serp-2 further demonstrated a greater affinity for binding to T-cells than CrmA upon flow cytometric analysis [40]. Of interest, CrmA has a greater inhibitory activity in vitro than Serp-2 for these targeted proteases, underscoring the differences in the in vitro and in vivo analyses of these unique virus-derived serpins.

Similar to studies with Serp-1, Serp-2 and CrmA have been tested as immune-modulating proteins in mouse models of transplant and carotid cuff compression injury. These studies are described below.

2-1B)-1.Atherosclerotic Plaque and Restenosis following Angioplasty Injury

In mouse models of atherosclerotic plaque growth and intimal hyperplasia after arterial injury (termed restenosis), both Serp-2 and CrmA were assessed for their efficacy in reducing plaque growth. In balloon angioplasty and aortic transplant models, Serp-2, but not CrmA nor Serp-2 RCL mutants, demonstrated statistically significant reductions in arterial plaque. Serp-2 treatment was also investigated by Bot and Biessen in hyperlipidemic ApoE^null^ mice following carotid cuff compression injury. In this model, yet again, Serp-2 demonstrated reduced plaque growth and macrophage invasion. Of interest, the systemic increase in aortic plaque, that was seen proximal to the carotid injury in the hyperlipidemic ApoE^null^ mouse model, demonstrated an attenuated inflammatory response with greater reductions in plaque in the aorta proximal to the site of carotid cuff compression injury, indicating a systemic reduction in plaque growth. CrmA, again, did not reduce inflammation nor plaque, further demonstrating the specific immune modulating effects of Serp-2 [40].

To examine the mechanisms for Serp-2-mediated reductions in arterial plaque growth, the signaling pathways were investigated. The in vitro analysis of Serp-2 and its effect on monocytes revealed decreased monocyte activation, as well as decreased membrane fluidity. Changes in membrane fluidity are seen in a variety of pathogenic states including hypercholesterolemia, hyperlipidemia, and atherosclerotic heart disease. Signaling pathway activation was also reduced by Serp-2, with detected decreases in WNT2, which is an important signaling molecule for cell-to-cell communication. The decreased atherogenic molecule expression of CD5, SELE, and VCAM1 was seen when monocytes were treated with Serp-2, to a greater extent than was seen with Serp-1, providing another molecular pathway for atherosclerotic plaque reduction [23,41].

2-1B)-1-2.Transplant Rejection2-1B)-1-2-1.Intimal Plaque Hyperplasia in Aortic Transplant

In aortic allograft transplants, Serp-2 reduced inflammation and intimal plaque development. Granzyme B deficiency in donor aortic allografts attenuated Serp-2 efficacy, reducing the Serp-2-mediated reduction in aortic plaque and intimal hyperplasia (thickening), and indicating Granzyme B as a central target for Serp-2-mediated anti-inflammatory and anti-atherogenic activity. In this aortic allograft model, Serp-2 but not CrmA reduced monocyte apoptosis without affecting the T-cells. Thus, in conclusion, multiple in vitro and in vivo studies have demonstrated that Serp-2 has the potential to inhibit arterial vascular disease progression through the immune-modulating inhibition of Granzyme B-dependent apoptosis [40].

2-1B)-1-2-2.Ischemia-Reperfusion Injury in Liver Transplant

Following liver transplantation, ischemia-reperfusion injury causes acute transplant rejection and allograft transplant loss. To model this, mice sustained 90 min of hepatic injury and were then treated with Serp-2 with comparison to a control. Serp-2 improved the survival along with a statistically significant reduction in liver enzyme (ALT) levels, infarct scar thickness, and hepatocyte necrosis [41].

2-1C).Orthopoxviral Serpins Spi1, Spi2 and Spi3

Orthopoxviruses include smallpox, cowpox, vaccinia, horsepox, camelpox, and monkeypox (now called Mpox). There is extensive shared activity and shared gene sequences between the cowpox virus and vaccinia virus genes referred to as CrmA, CrmB, and CrmC or Spi1, Spi2, and Spi3. The terms SPI1, 2, and 3 were initialy coined to stand for Serine Protease Inhibitors, but with ongoing research there has been some overlap in terminology for the term Spi used to refer to mammalian cellular serpins. These cowpox and vaccinia serpins are often referred to as interchangeable. Spi2 and CrmA share protease targets in the apoptotic pathways while Serp-1 and Spi3 share targets in the thrombotic and thrombolytic pathways [48,59,60]. There have been some interesting preclinical studies on these serpins; however, work with these serpin classes as potential protein treatments and remains in the early stages. These studies are described below.

2-1C)-1.Spi1 and Spi2


*Cancer: Potential for Oncolytic Therapeutic*


SPI-2 in combination with SPI-1 protects cells from alloreactive cytotoxic T lymphocyte (CTL)-induced killing via perforin and death receptor-dependent pathways in vitro. Spi-2, also termed CrmA, is a potent inhibitor of apoptosis [48]. Inhibitors of apoptosis can be considered as potential therapeutics for cancer therapy. Through antagonizing inhibitors of apoptosis, it is proposed immune cell responses to tumors can be improved. A similar approach designed to reduce immune cell apoptosis has been investigated as an oral apoptosis protein inhibitor in patients with advanced solid tumors [59].


*Viral Infections*


As noted, poxviral serpins can inhibit apoptosis during viral infection in vitro. Through the inhibition of apoptosis, Spi1 has potential as an antiviral therapeutic for poxviral infections by prolonging the viral antigen presentation to host dendritic cells in the immune response [48]. The inactivation of immunomodulating genes such as B13R (encoding Spi1) or B22R (Spi2) has been proposed as an approach to enhance the safety of vaccinia virus vaccines while maintaining a high level of immunogenicity in general vaccination [60].

2.1C)-2.Spi2/CrmA

CrmA is a member of the Ov-serpins clade, which lack typical, cleavable hydrophobic signal sequences, resulting in inefficient (and often undetectable) translocation and secretion. CrmA, similar to Serp-2, is a predominantly cytoplasmic serpin [Figure 2]. In the mouse aortic allograft transplant studies, both Serp-2 and CrmA were assessed for efficacy in reducing transplanted aortic inflammation. In this model, Serp-2 did significantly reduce transplant vasculitis, inflammation, and plaque growth; however, CrmA was not effective despite having greater in vitro inhibitory activity [40]. Thus, the question arises as to whether other protease targets may be involved in the efficacy demonstrated for Serp-2 in these transplant vascular models.


*Immunotherapeutics*


As noted above, Spi2, when transfected into cells, inhibits apoptosis through the targeting of Granzyme B (GrzmB) and caspases [40]. CrmA targets the serine protease GrzmB as well as the cysteine proteases caspases 1 and 8. SPI-2, in combination with SPI-1, protects cells from alloreactive CTL-induced killing via perforin and death receptor-dependent pathways in vitro. Serp-2 and CrmA thus bind and inhibit similar classes of serine and cysteine proteases, specifically Granzyme B and caspases 1 and 8 [40]. Turner and Moyer et al. examined the effects of swapping the RCL sequences of Serp-2 with CrmA. However, CrmA sequences, inserted into the Serp-2 protein, did not retain the same level of anti-apoptotic activity as seen with the wild type serpins [60,61,62].

Sindbis virus is an arthropodal (insect) virus of genome alphavirus. Sindbis is transmitted between vertebrate (bird) and invertebrate (mosquito) vectors. The virus itself induces classical features of apoptosis. In vivo, treatment with CrmA serpin from the cowpox virus inhibited sindbis virus-induced cell death (apoptosis) rather than impairing viral replication, improving the survival in infected mice. This is an early study of the use of a cowpox virus-derived inhibitor of caspase (CrmA) demonstrating improved outcomes during infection by an unrelated Sindbis virus [63].


*Neuroprotection*


Apoptosis can alter neuroexcitability in neurodegeneration disorders. In vitro studies have demonstrated that CrmA treatment was able to rescue neurons, reducing toxicity following a necrotic infection, even in conditions of low caspase activation and morphological apoptosis. CrmA apparently reduced the drop in mitochondrial potential and the reduction in ATP. This work suggests a potential therapeutic application for the cowpox virus-derived CrmA [49].


*Viral Infections*


Cytotoxic T lymphocytes (CTL) induce cell death after viral infection by the secretion of perforin/granzymes and Fas cell surface antigen. Proteinase activity is important in these CTL-mediated cell death pathways. SPI-2 inhibits the proteolytic activity of caspase 1 (Interleukin 1 beta, IL-1b, converting enzyme) and granzyme B. Cells infected with the orthopoxviruses cowpox and rabbitpox are resistant to cytolysis via these mechanisms. Mutation of the SPI-2 gene prevents virus inhibition of Fas-mediated cytolysis, allowing cell death to proceed and reduce viral replication. The mutation of both SPI-2 and SPI-1 is reported to completely reverse or block viral cytolysis.

The viral inhibition of perforin/granzyme-mediated cell killing was unaffected by mutation of SPI-2, and perforin-/granzyme-mediated killing was reduced when SPI-1 genes were inactivated. Thus, SPI-1 and SPI-2 together inhibit both cytolysis pathways, and also a common pathway enzyme, IL-1b convertase, which is associated with both perforin/granzyme and Fas-cell surface killing [50].


*Fulminant and Autoimmune Hepatitis*


Fulminant hepatitis can occur with severe viral infections, as well with chemical toxicity with a rapid progression of liver failure within days to weeks, causing massive liver parenchymal necrosis. Fas-mediated cytolysis and perforin-/granzyme-mediated killing have a role in this hyperimmune response in hepatitis. CrmA inhibits both Granzyme B and caspase activation. The injection of anti-Fas antibody into mice leads to death due to liver cell apoptosis. In mice with Adenovirus expression of higher levels of CrmA, apoptosis and hepatitis were dramatically reduced, with increased survival. Additionally, in in vitro analysis, active Caspase 3 was inhibited by the transfection of CrmA into mouse hepatocytes. Thus, CrmA may block immune-mediated apoptosis, as seen in autoimmune hepatitis [61]. Concanavalin-A (Con A)-induced hepatitis is a model for studying and mimicking human autoimmune hepatitis, with a massive hepatocyte apoptosis and CD11b+ leukocyte infiltration. This liver damage was also dramatically reduced in mice that expressed the CrmA gene, without effecting CD4+ cell responses. Mouse survival was also increased to 100% when compared to the control group [50,51].


*Detection of Resurgent Variola (Smallpox) Pandemic Infections*


One recent analysis of smallpox gene expression has projected that the detection of the expression of three smallpox genes may allow for the detection of Variola variants that are predictive of risk for recurrent smallpox epidemics. One gene that appears to have diagnostic and predictive potential is the CrmB serpin gene as expressed by variola [64].


*Ischemia Reperfusion following Myocardial Infarction*


Ischemia reperfusion injuries are also closely associated with apoptosis. To deliver extracellular CrmA into eukaryotic cells, a TAT transduction domain from HIV was fused onto the N-terminus of CrmA, allowing intracellular translocation. In this work, CrmA inhibited the intrinsic apoptotic pathway, inhibiting Caspase-8, Caspase-9, and Caspase-3 in vitro [52].

In vivo, 90% of mice survived a lethal dose of antibody to Fas when treated with CrmA [50]. To examine the extrinsic pathway in vivo, mice were treated with a lethal dose of doxorubicin in the control group, with another group receiving CrmA. A total of 40% of those who received TAT-crmA via intraperitoneal injection survived, whereas those without died within 31 days. TAT-crmA was also found to reduce myocardial infarct size by 40% in treated mice while preserving left ventricular systolic function. This study provides preclinical evidence of the potential use of CrmA in treating ischemia reperfusion injury in the setting of a myocardial infarction (MI) [53]. The temporal effects of CrmA were also examined in Doxorubicin-induced apoptotic myocardial injury and cardiomyopathy. The early, six-day survival was increased to 81% in CrmA-treated mice vs 38% in control mice. This effect disappeared by day 12, with similar levels of apoptosis and survival having been observed [53].

2.1C)-2-3.Spi32.1C)-2-3-1.Inflammation and Vascular Permeability

Spi3 is a cowpox viral protein. Spi3 complexes with and inhibits a variety of thrombolytics, including uPA, plasmin, and tPA, similarly to the Myxomaviral Serp-1 [Figure 2] or the mammalian serpins PAI-1 and NSP. Through the inhibition of thrombolytic proteases, Spi3 has potential for both research and the treatment of bleeding disorders such as hemophilia and vascular inflammation and permeability, albeit not yet proven. Hemagglutinin has been shown to retain Spi3 in the plasma membrane [54,55].

Both Spi3 (from cowpox virus) and Serp-1 (from myxomavirus) exhibit anti-inflammatory activity, inhibiting uPA, plasmin, and tPA. Both have arginine at the P1 site in the P1P1′ scissile bond in the RCL. Both also have the capacity to complex with thrombin or factor Xa. However, they are not functionally equivalent. Unexpectedly, when cowpox virus expresses Spi3 under the myxomavirus Serp-1 promoter, there is greater Spi3 secretion; thus, either the cowpox virus inhibits Spi3 through another mechanism, or myxoma enhances the secretion of Spi3 [65].

2-2).Herpesviral serpins

ORF1 (Open reading frame 1) in the murine gammaherpesvirus 68 (MHV68) encodes for a protein that contains amino acid sequences which are similar in homology to the poxvirus Serp-1 protein. The potential for therapeutic clinical applications of ORF1 has yet to be determined; however, this does present an example of a serpin-like molecule in Herpesviruses [66]. Herpes simplex virus-1 (HSV-1) DNA is immunostimulatory both in vivo and in vitro, promoting T helper (Th1) cell responses [67]. Given the immunomodulatory action of many serpins, it is reasonable to predict that other viral serpins will be discovered with the capacity to combat antiviral immune responses. As has already been studied, the Myxomaviral Serp-1 protein improved survival and reduced lung and vascular inflammation in MHV68 mouse gamma herpesviral infection in interferon gamma receptor-deficient mouse models. These new herpes virus-derived serpins may also provide a new source of serpins with the potential for therapeutic approaches for diseases with chronic inflammatory responses, and perhaps keratitis.

2-3).Plant Viral Serpins

Plant viruses are believed to be safe for administration in humans, being from different kingdoms. Plant viruses are used to safely present epitopes to create novel vaccines. Recently, several plant viruses have demonstrated immunogenicity, inducing IgG titers in mice. Tobacco Mosaic Virus (TMV) and spherical particles from TMV induced immunogenicity with low self IgG titers. Thus, viruses from plants can be investigated as a potential safe adjuvant for novel vaccines. Furthermore, this study demonstrates that plant viruses have an immunogenic effect [68].

One such immunogenic effect may involve immunomodulating serpins. Several plant serpins have been identified following viral infection. Their function remains obscure. Rice-stripe virus is one of the most destructive viruses of rice, and its microarray expression has identified a Serpin named Serpin-5, a possible pathogen resistance protein. Its biological significance remains unknown [69].

2-4).Arthropod Serpins and Athropod-Viral Serpins—Baculovirus Serpins

### 2.2. Nucleopolyhedroviral Serpins

Baculoviruses are large DNA viruses that predominantly infect Lepidoptera (moths and butterflies). The Hesp018 protein is a functional cross-class serpin with inhibitory activity against serine and cysteine proteinases. Hesp018 is the first viral serpin homologue to be characterized outside of the chordopoxviruses and is encoded as a baculovirus gene [70]. Serpin 4 from baculovirus inhibits PPO activation and melanization in the Asian Corn Borer, Ostrinia furnacalis (Guenee), forming covalent complexes with serine proteases. The baculoviral Serpin 4, again, demonstrates the conservation of a Serpin suicide-inhibitory mechanism of action [70].

Baculovirus is a virus which commonly attacks the insect H. armigera. Melanization is an insect defense mechanism that is regulated by serpins. Proteolytic activation of prophenoloxidase (PPO), as for the RSV (Rice-stripe virus), kills baculovirus. In H. armigera, baculovirus downregulates the protein levels in the PPO cascade and innate immune responses, but upregulates serpin5 and serpin9 expression in the hemolymph of insects. This is similar to the effects of the RSV on host-insect serpin pathways, a secondary effect of viruses on the insect expression of serpins. The inhibition of these serpins increases baculoviral infection. This suggests that the insect virus baculovirus has evolved a mechanism to combat the insect host immune response to the virus, again illustrating that serpins are important for viral immunomodulation in the host [66]. This response represents a virus-induced alteration in the host serpin response where the virus uses the host serpins as part of a defense mechanism.

### 2.3. Rice Stripe Virus Serpins (RSV)

As noted, a serpin sequence has been identified in the Rice-stripe virus, RSV. Furthermore, viruses are reported to hijack host-insect serpins. Rice gall dwarf virus hijacks the sperm protein HongrES1 in order to facilitate its spread, helping another virus to spread in the host, the symbiotic virus Recilia dorsalis filamentous virus. The activation of this serpin also facilitates decreased melanization and allows arbovirus host to spread. Thus, different viruses have been shown to act cooperatively through different kingdoms to help each other spread via these highly conserved serpin pathways [69].

RSV, a highly pathogenic virus for rice plants, is transmitted to the rice plant in a small planthopper insect. In one study, the transcription of seven serpins, termed Lsserpin1-7, has been detected in the insect. Phenoloxidase (PO) activity is activated by serine proteases in the insect vector and is one of the immune responses that are mounted by the host insect against viral infections. PPO activity is suppressed by the upregulation of these insect serpins during RSV infection, and the suppression of PPO can be up to 60% after upregulation (increased expression) of these serpins. The knock out of several of these serpins resulted in dramatically increased PO inhibitor activity, increasing innate immunity in the insect hemolymph [69]. Thus, RSV can induce the increased expression of serpins in the plant hopper insect, representing another virus-mediated activity that is related to altered insect host serpin expression.

## 3. Modifying Viral Serpins—New Therapeutic Constructs

Of great interest, recent work has illustrated the capacity for mammalian serpins to be modified such that these serpins can provide the selective or targeted blockade of coagulation pathways. The RCL of a human variant of A1AT, called A1AT Pittsburg, has been mutated to a KRK sequence, converting it into a specific inhibitor of activated protein C, termed SerpinPC. The results of a Phase 2 clinical trial presented at a hematology congress demonstrated that SerpinPC was well-tolerated and reduced bleeding in persons with severe haemophilia [71]. These serpins are in development as new approaches to reduce the excess activation of serpin pathways that can increase bleeding in hemophilia. Thus, viral serpins, as for mammalian serpins, can be modified to alter protease target specificity. The MyxV Serp-1 gene sequence has been mutated in prior work wherein Serp-1 anti-inflammatory activity was either lost or led to excess inflammation with aneurysm development [42]. Several RCL peptides derived from the Serp-1 RCL sequence have also been developed and tested in mouse models of vasculitis [43]. While wild type (WT) Serp-1 lost efficacy in a MHV68 model of vascular disease and lung hemorrhage after antibiotic suppression of the gut microbiome, one such Serp-1 RCL peptide retained anti-inflammatory activity in this model [72]. The modulation of pathways targeted by these RCL mutants of Serp-1 and/or the RCL peptides may provide interesting approaches to understanding protease pathways, as well as interaction with the microbiota that are central to disease development. Serpins may thus provide a mechanism utilized by viruses to alter the gut microbiome and their effects on host immune responses.

## 4. Conclusions

Given the widespread distribution of serpins in biology, reflecting the adaptability of the serpin protein structure and mechanisms of inhibition to different physiological contexts, there is great interest in their potential therapeutic benefit when used as drugs. Viral serpins have been shown to modulate the immune response with a therapeutic benefit in over 30 different diseases of inflammation with minimal if any adverse effects. This review highlights the known viral serpins and disease models that have been studied to date. Given what is presently known about serpins and their widespread studied therapeutic benefits, it is reasonable to continue translational and clinical research on serpin therapy on these disease states. Given the extensive variety of serine proteases that are inhibited by viral serpins, dedicated modifications of these serpins that are designed to target specific target proteases may provide new targeted treatment approaches, similar to those that have recently been developed for modified serpin treatment in hemophilia [2]. One might also consider viruses as a rich source of immune-modulating therapeutics for many other classes of proteins that target different immune pathways. In addition to serpins, chemokine modulators, growth factors, and viral cytokines such as vIL-10 have been studied as virus-derived immune-coagulopathic and apoptosis-modulating biologics, representing a new class of virus-derived therapeutics [43,58,72,73,74].

## Figures and Tables

**Figure 1 biomolecules-13-01393-f001:**
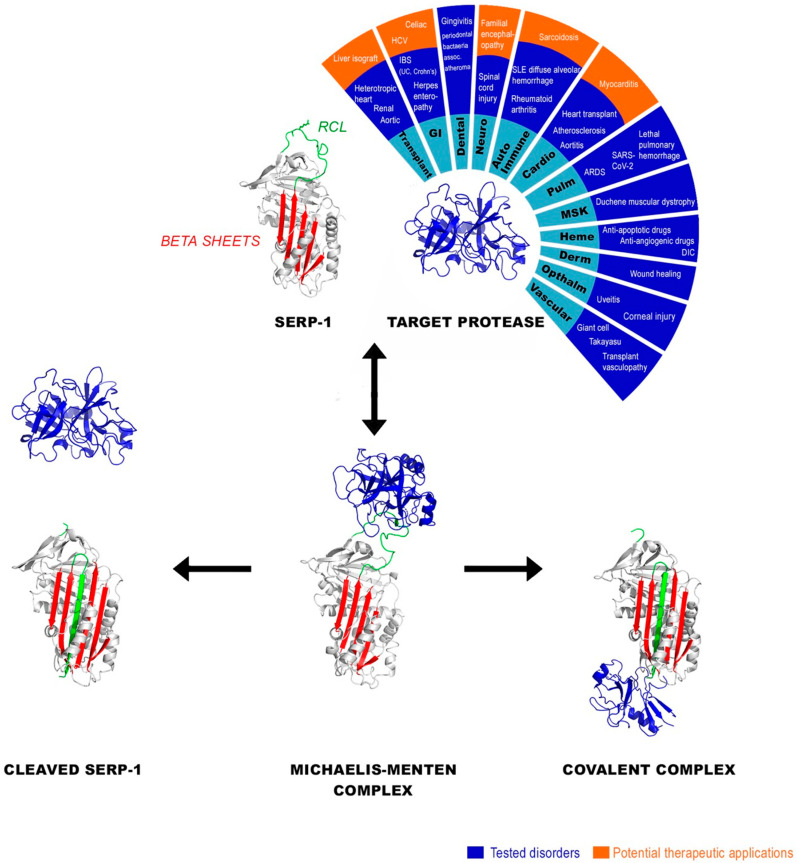
Proposed mechanisms of Serp-1 with prior preclinical and clinical studies examining the potential for virus-derived serpin protein therapeutics. Serp-1 is a multifunctional serine protease inhibitor, a serpin. The reactive center loop, RCL, of a serpin is cleaved by an active protease with subsequent formation of a covalent bond, forming an inactive serpin–protease complex, with the cleaved RCL forming an additional strand in the A beta sheet of the serpin, a form of suicide inhibition. Serpins are inhibitors, not proteases, with the capacity to (1) focus inhibition at sites of protease activity, (2) bind multiple targets, and (3) be modified to allow fine tuning of inhibitory functions. Serp-1 has potential as a therapeutic for a wide array of inflammatory and coagulopathic disorders. Orange—postulated efficacy, blue—prior preclinical and/or clinical efficacy.

**Figure 2 biomolecules-13-01393-f002:**
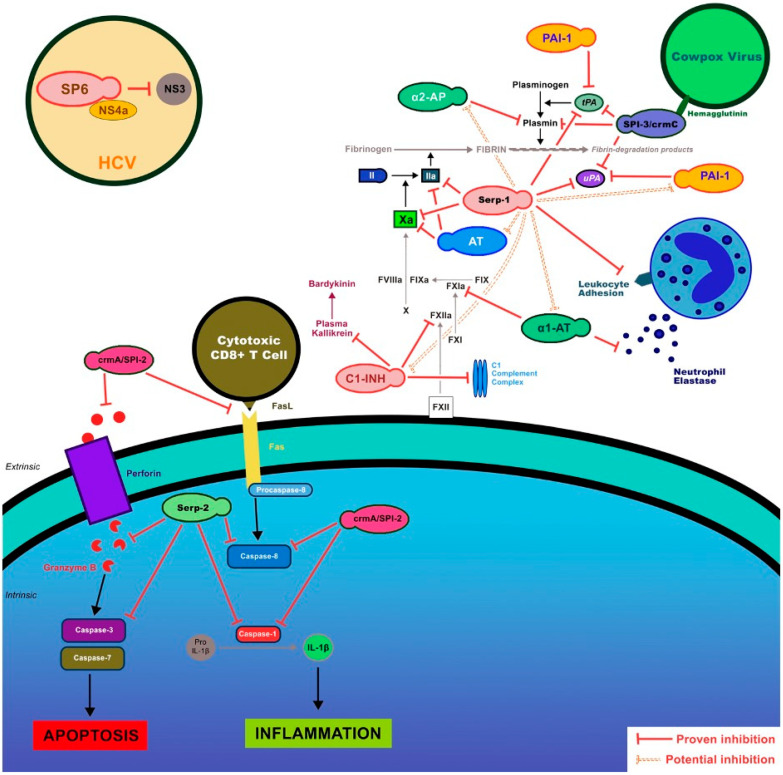
Immune-modulating mechanisms of myxoma poxvirus-derived serpins. Serp-1 operates predominantly extracellularly to inhibit thrombotic, thrombolytic, and complement proteases, reducing leukocyte recruitment. Serp-2 and CrmA operate predominantly intracellularly to inhibit a variety of pro-inflammatory and pro-apoptotic factors. However, Serp-2 has immune-modulating functions when given systemically, via an extracellular or systemic infusion. SPI-1and SPI-2/ CrmA are multi-pathway inhibitors of apoptosis. SPI-3 targets coagulation factors extracellularly. A1AT—alpha 1 anti-trypsin, A2AP—alpha 2 antiplasmin, AT—anti-thrombin, C1Inh—C1 complement esterase inhibitor 1, CRM—Cytokine response modifier, IL-1b—interleukin 1beta, PAI-1—plasminogen activator inhibitor -1, SPI—serine protease inhibitor, tP—tissue type plasminogen activator, uPA—urokinase-type plasminogen activator, II—factor II, X—factor X.

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
