# Peer review of "Viral SERPINS—A Family of Highly Potent Immune-Modulating Therapeutic Proteins"

_biomolecules, 2023, doi:10.3390/biom13091393_

Round 1

Reviewer 1 Report

In this manuscript, the authors presented an overview on known viral serpins, prior studies and potential for applications as biologic reagents for immune, coagulation and apoptosis modulatory therapeutics. The review is very comprehensive and may serve as an important reference for people interested in this field.

Comments:

# The resolution of figure 2 needs to be improved.

# Line 271-272, it is not very useful to only show p-values without showing actual values, consider removing the p-values.

# The authors are recommended to re-check for unnecessary use of comma, e.g. line 281.

# Information incomplete for reference#7.

# As the authors nicely presented, viral SERPINS have been found to inhibit a wide variety of serine proteases. For example, Serp1 binds thrombolytic and thrombotic proteases (tPA, uPA, plasmin, fXa and thrombin (in the presence of heparin) as well as complement proteases. This may also mean potential off-target side effects when used to treat certain diseases. In this regard, dedicated modifications of serpins for better specificity towards certain proteases maybe needed. Could the authors discuss a bit more on this, maybe in the final part of the review?

Author Response

Reviewer 1

General comments –

In this manuscript, the authors presented an overview on known viral serpins, prior studies and potential for applications as biologic reagents for immune, coagulation and apoptosis modulatory therapeutics. The review is very comprehensive and may serve as an important reference for people interested in this field.

            Response – We thank the reviewer for these supportive comments.

Comments:

# The resolution of figure 2 needs to be improved.

Response – We thank the reviewer and have now revised Figure 2 to improve the resolution and also provided definitions for all abbreviations used in the figure legend.

# Line 271-272, it is not very useful to only show p-values without showing actual values, consider removing the p-values.

            Response – We thank the reviewer for this valid comment and we have removed the p values.

# The authors are recommended to re-check for unnecessary use of comma, e.g. line 281.

            Response  -We have extensively proofed and corrected the review.

# Information incomplete for reference#7.

            Response – The references have now been proofed

# As the authors nicely presented, viral SERPINS have been found to inhibit a wide variety of serine proteases. For example, Serp1 binds thrombolytic and thrombotic proteases (tPA, uPA, plasmin, fXa and thrombin (in the presence of heparin) as well as complement proteases. This may also mean potential off-target side effects when used to treat certain diseases. In this regard, dedicated modifications of serpins for better specificity towards certain proteases maybe needed. Could the authors discuss a bit more on this, maybe in the final part of the review?

Response – We thank the reviewer for this excellent comment. We have now provided a brief commentary on the previously developed mutations of the wild type (WT Serp-1) RCL as well as peptides developed and derived from the Serp-1 RCL  and their modified functions when examined in mouse models. Future work examining altered inhibition of select proteases will be of great interest.

These revisions are as follows, lines 824- 842

Modifying Viral Serpins – New Therapeutic Constructs

Of great interest, recent work has illustrated the capacity for mammalian serpins to be modified such that these serpins can provide selective or targeted blockade of coagulation pathways. The RCL of a human variant of A1AT called A1AT Pittsburg has been mutated to a KRK sequence converting it into a specific inhibitor of activated protein C, termed SerpinPC. Results in a Phase 2 clinical trial presented at a hematology congress demonstrated that SerpinPC was well-tolerated and reduced bleeding in persons with severe haemophilia [72]. These serpins are in development as new approaches to reduce excess activation of serpin pathways that can increase bleeding in hemophilia. Thus viral serpins, as for mammalian serpins, can be modified to alter protease target specificity. The MyxV Serp-1 gene sequence has been mutated in prior work wherein Serp-1 anti-inflammatory activity was either lost or led to excess inflammation with aneurysm development (73). Several RCL peptides derived from the Serp-1 RCL sequence have also been developed and tested in mouse models of vasculitis (74).  While wild type (WT) Serp-1 lost efficacy in a MHV68 model of vascular disease and lung hemorrhage after antibiotic suppression of the gut microbiome, one such Serp-1 RCL peptide retained anti-inflammatory activity in this model [75]. The modulation of pathways targeted by these RCL mutants of Serp-1 and / or the RCL peptides may provide interesting approaches to understanding protease pathways, as well as interaction with the microbiota central to disease development. Serpins may thus provide a mechanism utilized by viruses to alter the gut microbiome and effects on host immune responses.”

Reviewer 2 Report

This review work by Varloly et al. presents an exhaustive revision of the applications of viral serpins as therapeutical molecules. While this review has the merit of being comprehensive, it would benefit from a careful revision of the writing style and grammatical correctness, taking care to use commas when needed, avoiding unnecessary breaking down into paragraphs, and specially avoiding reiterations. The following are important points that need to be addressed:

-       Lines 86, 87 and 88: this description of serpin polymer formation is not valid anymore, and no reference is given for the mechanism described. The authors should report the findings about the C-terminal domain swap mechanism for polymer formation, as seen in vitro (Yamasaki et al., 2011, EMBO J) and even more relevant for polymers of alpha-1 antitrypsin isolated from the liver (Faull and Elliston et al., 2020, Science Advances).

-       Lines 99, 100 and 101: the description of the phenotype-genotype correlation observed for mutant variants of neuroserpin that cause FENIB is very poor. I suggest rewriting this sentence and giving the original reference for this work (Davis et al., 2002, Lancet).

-       Line 103, 104: alpha-1 antitrypsin deficiency must be mentioned in this sentence, as it is the most common, better characterized serpinopathy.

-       Figure 2: this figure, although useful, is difficult to read, many of the names and some boxes are too small, the acronyms are not defined in the figure legend, the last sentence in the legend seems incomplete. Also, the quality of the figure is not high enough and many writings appear out of focus. I suggest revising the figure carefully.

Some of the minor edits that the text needs:

All text: no need to use capital S for Serpins, better use serpins; in general, no need to use capitals for protein names and other common names, even when introducing an acronym; indeed, the use in this manuscript is inconsistent (sometimes yes, sometimes not); apply this comment to all the text (another example: Alpha-1 Antitrypsin in line 91).

All text: lattin names for species should be correctly written, using italics, with first capital letter for the gender and non-capital for the species; also, be consistent with the gender: always complete or always abbreviated; for example, in line 40 change homo sapiens to Homo sapiens.

Figure 1: define all the abbreviations used in the protease function scheme in the legend (for example GI, Pulm, etc).

Figure 1: Michaelis-Menten complex is not correct, it is a Michaelis complex.

Line 60: cross-class (dash needed in many other places across the text).

Line 109: The sentence repeats twice the same concept, revise it to avoid reiteration; in general, this paragraph from line 109 to 126 has repetitions and is difficult to read.

Line 158: delete the unnecessary full stop.

Lines 132 and 133: the sentence is cut at functions, needs rewriting.

Paragraph starting at line 179: Cowpox again written with capital C, while horsepox is not; I strongly suggest avoiding excessive use of capitals.

Line 224: I think figure 2 would be better cited at line 212.

Line 303: the sentence ‘local application of P. Gingivitis, caused by a pathogenic oral bacterium,…’ does not make sense.

Line 427: it is not correct to use ug instead of µg.

Line 487: I assume WTSerp-1 to mean wild type Serp-1, but it is not defined and is used only once, so I suggest to use wild type Serp-1 instead.

The manuscript would benefit from a careful revision of the writing style and grammatical correctness, taking care to use commas when needed, avoiding unnecessary breaking down into paragraphs, and specially avoiding reiterations and unnecessary experimental details that make the reading a bit tedious.

Author Response

Reviewers comments -

Reviewer 2

Comments and Suggestions for Authors

This review work by Varloly et al. presents an exhaustive revision of the applications of viral serpins as therapeutical molecules. While this review has the merit of being comprehensive, it would benefit from a careful revision of the writing style and grammatical correctness, taking care to use commas when needed, avoiding unnecessary breaking down into paragraphs, and specially avoiding reiterations. The following are important points that need to be addressed:

Response to general comment -  We thank the reviewer for this comment and we have carefully revised this review to modify the writing style. Please see the revised manuscript.

Comment 1.

“Lines 86, 87 and 88: this description of serpin polymer formation is not valid anymore, and no reference is given for the mechanism described. The authors should report the findings about the C-terminal domain swap mechanism for polymer formation, as seen in vitro (Yamasaki et al., 2011, EMBO J) and even more relevant for polymers of alpha-1 antitrypsin isolated from the liver (Faull and Elliston et al., 2020, Science Advances).”

Response  - We thank the reviewer for this comment and we have revised the introduction to include descriptions of both currently accepted mechanisms for serpin polymerization, the loop-sheet and the domain swap mechanisms of polymerization. We agree with incorporating a review of both mechanisms and have provided this, however current literature does not indicate that only one mechanism is viable. Both mechanisms have been extensively discussed. It has been suggested that many differing areas of mutated serpins may interact and thus we have provided a brief discussion of these two accepted mechanisms. The modified section in the Introduction now reads as follows; lines 86-109

The impact of serpins as regulatory proteins that maintain normal physiological functions is highlighted by the profound effects of genetic mutations in serpins that cause a class of disease known as a serpinopathy. These mutations lead in some cases to a loss of inhibitory function and in other cases a gain of function. In some of these conditions, deposition of serpin aggregates and protein misfolding is associated with tissue damage and leads to a deficiency state which prevents effective inhibition of target proteases. For neuroserpin (SERPINI1) and alpha 1 antitrypsin (A1AT, SERPINA1), this aggregation is mediated by a repeating intermolecular contact that generates linear polymers ‘beads-on-a-string’ morphology [6,7]. In A1AT deficiency, polymer formation in the liver causes liver damage (cirrhosis) as well as lack of effective inhibition of lung neutrophil elastase causing severe emphysema (lung damage) [8,9]. For neuroserpin (SERPINI1), polymer formation in the central nervous system is the basis for a hereditary dementia termed Familial Encephalopathy with Neuroserpin Inclusion Bodies (FENIB). Notably, the onset and severity of neurodegenerative disease was associated with rate and magnitude of neuronal protein aggregation [7,10,11]. 

As polymers share a similar increase in stability to the RCL-inserted, protease-cleaved form of the protein, and polymerization is prevented by pre-incubation with a peptide mimetic of the RCL, it was proposed that polymers form by insertion of the RCL from one serpin into the β-sheet A of an adjacent serpin. This is often referred to as the ‘loop-sheet model’[1-5]. The feasibility of an alternative mechanism in which polymerization is mediated by a domain swap of the entire 4kDa C-terminus of the protein has been demonstrated crystallographically using a double cysteine mutant of alpha-1-antitrypsin supported by disulfide trapping experiments [8,12]. Subsequent structural characterization of polymers extracted from patient liver strongly supports this as the form present in vivo [7,13].”

Comments 2 and 3 - Lines 99, 100 and 101: the description of the phenotype-genotype correlation observed for mutant variants of neuroserpin that cause FENIB is very poor. I suggest rewriting this sentence and giving the original reference for this work (Davis et al., 2002, Lancet). 

-       Line 103, 104: alpha-1 antitrypsin deficiency must be mentioned in this sentence, as it is the most common, better characterized serpinopathy.

Response – We thank the reviewer for these comments and these sections have been revised as follows. Please see the revised sections below lines 91-99

. For neuroserpin (SERPINI1) and alpha 1 antitrypsin (A1AT, SERPINA1), this aggregation is mediated by a repeating intermolecular contact that generates linear polymers ‘beads-on-a-string’ morphology [6,7]. In A1AT deficiency, polymer formation in the liver causes liver damage (cirrhosis) as well as lack of effective inhibition of lung neutrophil elastase causing severe emphysema (lung damage) [8,9]. For neuroserpin (SERPINI1), polymer formation in the central nervous system is the basis for a hereditary dementia termed Familial Encephalopathy with Neuroserpin Inclusion Bodies (FENIB). Notably, the onset and severity of neurodegenerative disease was associated with rate and magnitude of neuronal protein aggregation [7,10,11].” 

Comment 4-      Figure 2: this figure, although useful, is difficult to read, many of the names and some boxes are too small, the acronyms are not defined in the figure legend, the last sentence in the legend seems incomplete. Also, the quality of the figure is not high enough and many writings appear out of focus. I suggest revising the figure carefully.

Response  - We do thank the reviewer for this suggestion – We have carefully revised the figure as well as providing definitions for the acronyms used. Please see the revised Figure 2.

Minor edits - Some of the minor edits that the text needs:

Edit 1 - All text: no need to use capital S for Serpins, better use serpins; in general, no need to use capitals for protein names and other common names, even when introducing an acronym; indeed, the use in this manuscript is inconsistent (sometimes yes, sometimes not); apply this comment to all the text (another example: Alpha-1 Antitrypsin in line 91).

Response – Thank you - These inconsistencies have now been corrected throughout the manuscript.

Edit 2 - All text: lattin names for species should be correctly written, using italics, with first capital letter for the gender and non-capital for the species; also, be consistent with the gender: always complete or always abbreviated; for example, in line 40 change homo sapiens to Homo sapiens.

Response – These inconsistencies have been corrected throughout the review manuscript.

Figure 1: define all the abbreviations used in the protease function scheme in the legend (for example GI, Pulm, etc).

Response – these abbreviations are now defined.

Edit 3 - Figure 1: Michaelis-Menten complex is not correct, it is a Michaelis complex.

We have reviewed the term Michaelis Menten and it would appear that the term refers to the two researchers Michaelis and Menten who first described protease substrate interactions. Thus we would respectfully note that the use of this term is acceptable.

Edit 4 - Line 60: cross-class (dash needed in many other places across the text).

            Response – now corrected for consistency

Line 109: The sentence repeats twice the same concept, revise it to avoid reiteration; in general, this paragraph from line 109 to 126 has repetitions and is difficult to read.

            Response -the repetitions have been removed and the paragraph improved.

Line 158: delete the unnecessary full stop.

            Response – this is corrected.

Lines 132 and 133: the sentence is cut at functions, needs rewriting.

Response – this is corrected.

Paragraph starting at line 179: Cowpox again written with capital C, while horsepox is not; I strongly suggest avoiding excessive use of capitals.

            Response – We have now made the use of capitols consistent throughout the tecxt.

Line 224: I think figure 2 would be better cited at line 212.

Line 303: the sentence ‘local application of P. Gingivitis, caused by a pathogenic oral bacterium,…’ does not make sense.

            Response – We thank  the reviewer for this edit. This is now corrected -

Line 427: it is not correct to use ug instead of µg.

            Response – Thank you and we agree – this is corrected.

Line 487: I assume WTSerp-1 to mean wild type Serp-1, but it is not defined and is used only once, so I suggest to use wild type Serp-1 instead.

            Response – We thank the reviewer and have defined WT in the text.
